# The Inability of Spotted Lanternfly (*Lycorma delicatula*) to Vector a Plant Pathogen between its Preferred Host, *Ailanthus altissima*, in a Laboratory Setting

**DOI:** 10.3390/insects11080515

**Published:** 2020-08-09

**Authors:** Rachel K. Brooks, Ashley Toland, Andrew C. Dechaine, Thomas McAvoy, Scott Salom

**Affiliations:** 1Forest Health and Resiliency Division, Washington Department of Natural Resources, Olympia, WA 98504, USA; 2Integrated Pest Prevention and Management Program, Oregon Department of Agriculture, Salem, OR 97301, USA; ashleyat@vt.edu; 3Department of Entomology, Virginia Tech, Blacksburg, VA 24061, USA; dechaine@vt.edu (A.C.D.); tmcavoy@vt.edu (T.M.); salom@vt.edu (S.S.)

**Keywords:** *Verticillium nonalfalfae*, biological control, biopesticide, tree-of-heaven, invasive species management

## Abstract

**Simple Summary:**

The invasive and accidently introduced insect, the spotted lanternfly, is spreading rapidly and becoming abundant in the mid-Atlantic region of the USA. Though this insect prefers to feed on the also invasive tree-of-heaven, its ability to feed on other native and crop plant species is concerning, and therefore eradication and control efforts are underway. These efforts include targeting the difficult to control tree-of-heaven for removal. Recently, researchers have found that a naturally occurring fungus effectively kills the tree-of-heaven and work towards making this fungus publically available is ongoing. Therefore, we tested whether the spotted lanternfly is capable of spreading the pathogen between symptomatic fungus-inoculated tree-of-heaven seedlings or plant material to healthy tree-of-heaven seedlings in a controlled laboratory setting. In these conditions, we found no evidence that this transmission occurred. This included monitoring the seedlings for symptoms and sampling the seedlings and the insects for the fungus. This lack of transmission may indicate that the spotted lanternfly cannot help spread this fungus to other tree-of-heaven.

**Abstract:**

With the recent introduction of the non-native spotted lanternfly (*Lycorma delicatula*) to the USA, research and concern regarding this insect is increasing. Though *L. delicatula* is able to feed on many different plant species, its preference for the invasive tree-of-heaven (*Ailanthus altissima*) is apparent, especially during its later life stage. Therefore, management focused on *A. altissima* control to help limit *L. delicatula* establishment and population growth has become popular. Unfortunately, the control of *A. altissima* is difficult. *Verticillium nonalfalfae*, a naturally occurring vascular-wilt pathogen, has recently received attention as a potential biological control agent. Therefore, we studied if *L. delicatula* fourth instars or adults could vector *V. nonalfalfae* from infected *A. altissima* material to healthy *A. altissima* seedlings in a laboratory setting. We were unable to re-isolate *V. nonalfalfae* from the 45 *A. altissima* seedlings or from the 225 *L. delicatula* utilized in this experiment. We therefore, found no support that *L. delicatula* could effectively vector this pathogen between *A. altissima* in laboratory conditions. Since *L.*
*delicatula*’s ability to vector *V. nonalfalfae* has implications for the dissemination of both this beneficial biological control and other similar unwanted plant pathogens, future research is needed to confirm these findings in a field setting.

## 1. Introduction

The recent accidental introduction and rapid spread of the spotted lanternfly, *Lycorma delicatula* (White) (Hemiptera: Fulgoridae), from its native range in China to Pennsylvania, US is a cause of much concern [1,2,3]. With economic impacts that are just beginning to be understood, it is likely that this hard to control insect will negatively impact vineyards, orchards, ornamental plants, timber trees, and residential properties [4,5,6,7].

*Lycorma delicatula*, an aggressive phloem feeder, has a large host range of more than 70 plant species [5,6]. However, studies from South Korea and the US suggest that the non-native tree-of-heaven (*Ailanthus altissima* (Miller) Swingle) is its preferred host. This is especially apparent as *L. delicatula*’s host range narrows during its later life stages [1,3,4,5,6,7]. The preference for *A. altissima* may be driven by the tree’s high sugar content [4] and *L. delicatula* ability to sequester the tree’s quassinoids to use as defense chemicals [7]. This relationship, and whether *A. altissima* is required for *L. delicatula* development, is the focus of ongoing research. 

Similar to *L. delicatula*, *A. altissima* is native to China, has invaded much of the US, and causes a wide range of negative impacts [8]. *Ailanthus altissima* is difficult to control using mechanical and chemical methods [9,10], and has therefore been the target of a variety of biological control efforts, such as the not-yet-released Chinese weevil *Eucryptorrhynchus brandti* (Harold) (Coleoptera: Curculionidae) [11,12]. More recently, a very promising naturally occurring fungus found killing *A. altissima* in Pennsylvania (*Verticillium nonalfalfae* Inderb. (formerly *Verticillium albo-atrum* Reinke and Berthold) has also been the subject of biological control [13,14,15]. Movement of this vascular wilt pathogen within a stand through functional root grafts is rapid and can result in the local mortality of hundreds of *A. altissima* trees per year [15,16]. Little is known about the spread of the fungus over longer distances between stands, despite the fact that *V. nonalfalfae* has been found killing *A. altissima* in three different states [13,17,18].

Since insects have the potential to vector plant pathogens, they can influence the large-scale dissemination of both beneficial biological controls and harmful plant pathogens. With regards to *V. nonalfalfae, E. brandti* has previously been shown in laboratory settings to vector this pathogen between *A. altissima* [19], but no research regarding *L. delicatula* as a plant pathogen vector has yet been published. Since the later instars and adults of *L. delicatula* are thought to disperse long distances and prefer *A. altissima* as a host [2], *L. delicatula* could play a significant role in disseminating *V. nonalfalfae* between *A. altissima* stands. Confirming the ability of *L. delicatula* to vector this proposed biological control agent would help land managers better predict the consequences of *L. delicatula* invasion and manage *A. altissima* more effectively.

The objective of this research was to determine if *L. delicatula* fourth instar nymphs or adults could transfer *V. nonalfalfae* from symptomatic *A. altissima* tissues to healthy *A. altissima* seedlings in a laboratory setting. We hypothesized that *L. delicatula* would be able to act as an effective vector of *V. nonalfalfae*. 

## 2. Materials and Methods

### 2.1. Overview

Overall, *L. delicatula* fourth instars or adults were allowed to feed for 48 h on either symptomatic of asymptomatic *A. altissima* plant material (either seedlings or logs) before being transferred to healthy *A. altissima* seedlings where they fed for 72 h. These seedlings were then allowed to grow insect free for an additional 3 weeks before being assessed for Verticillium wilt-like symptoms and destructively sampled for *V. nonalfalfae* isolation. Additionally, at the conclusion of the second feeding period, *L. delicatula* were washed in sterile water and the wash water was plated to determine if any viable *V. nonalfalfae* was present on the outside of the insects (Figure 1 and Figure 2).

During the entirety of this research, all permit requirements were followed (Virginia Department of Agriculture and Consumer Services Office of Plant Industry Service Spotted Lanternfly Permit # VASLF-19030 and USDA Animal and Plant Health Inspection Service, Plant Protection and Quarantine P526P-18-02138). Work conducted in the Beneficial Insects Containment Facility #62 at the Virginia Tech Prices Fork Research Center, (4057 Prices Fork Road, Blacksburg, VA, USA, here after “Quarantine Lab”) followed the facility’s approved Standard Operating Procedure guidelines. In addition, any handling of plant material and insects that had been in contact with *V. nonalfalfae* was done using sterile techniques.

### 2.2. Verticillium nonalfalfae Inoculum

Cultures of *Verticillium nonalfalfae* VnAa200/NRRL66918 [18] were grown on prune extract agar amended with streptomycin sulfate and neomycin sulfate (PEA + SN) [20] starting in mid-May 2019. After 2–5 weeks of growth, sterile 0.1% peptone in water was used to suspend *V. nonalfalfae* conidia at 10^7^ conidia ml^−1^ for immediate use as an inoculum [14,15]. The germination rate of the inoculum was confirmed to be over 75% both before and after use by plating a subsample of the suspension on water agar and assessing germination after 24 h.

### 2.3. Ailanthus altissima Seedlng Source

*Ailanthus altissima* seedlings were grown from seeds collected on 17 January 2019 in Blacksburg, VA and planted in trays filled with potting soil (Miracle-Gro^®^ Potting Mix 0.21-0.11-0.16, ScottsMiracle-Gro, Marysville, OH USA) on 29 January, 6 February, and 26 February 2019. Germinated seedlings were individually transferred to 3.8 L (1 gal) pots on 17 April 2019 and watered as needed.

To ensure that symptomatic and control seedlings would be present as needed throughout this experiment, these seedlings were inoculated with either *V. nonalfalfae* or a control on four different days (27 May, 5 June, 21 June, and 5 July 2019). At inoculation, a 0.1 ml *V. nonalfalfae* inoculum or a sterile peptone and water control was inoculated into a seedling’s xylem 1–2 cm above the soil line using a sterile syringe (SlipTip^TM^ 0.5 mm × 25 mm Insulin Syringe, Becton Dickinson and Co., Franklin Lakes, NJ, USA). Inoculated seedlings were kept at Virginia Tech’s Washington Street Greenhouse (Blacksburg, VA, USA) and watered as needed. When required for the experiment, seedlings were transferred to the Quarantine Lab and placed on clean 20.3 cm (8 in) clear plastic saucers (Dotchi Garden^®^, Miami, FL, USA) under grow lights at 25 °C, 14 light: 10 dark [19].

### 2.4. Ailanthus altissima Mature Tree Source

Mature trees were obtained from a 15-year old *A. altissima* stand that had regenerated within a previous clear-cut area at Virginia Tech’s Shenandoah Valley Agricultural and Research Center in Raphine, VA. This regeneration stand contained patches of healthy *A. altissima* and symptomatic *A. altissima* artificially inoculated during a separate field experiment [21]. *Ailanthus altissima* trees selected for harvest were either asymptomatic and outside of symptomatic areas or symptomatic (majority of canopy symptomatic and xylem stained) and within previously *V. nonalfalfae* artificially inoculated areas. *Ailanthus altissima* were felled on 30 August 2019 and cut into 35 ± 5 cm logs that ranged from 4 to 6 cm in diameter. The bottom of each log was then re-cut and immediately placed into separate 18.9 L (5 gallon) buckets containing 10 cm of water. When needed for the experiment, logs were removed from their buckets and placed individually into 0.95 L (32 oz) Deli Food Containers (Comfy Package) containing 5 cm of water and secured with Parafilm (Fisher, 13-374-10). 

### 2.5. Collecting L. delicatula

*Lycorma delicatula* were field collected in Winchester, VA (39.188, −78.131) when appropriate numbers of the desired life stages were reported. In total, at least 100 insects were collected during each collection event, allowing for a 25% mortality rate during transportation. Fourth instars were collected on 3 July 2019 and adults were collected on 25 July and 20 August 2019. 

During field collections, groups of ten insects were placed into hard-plastic containers (Square PET Jars, General Bottle Supply, Los Angeles, CA, USA), which were then placed on ice in hard-plastic coolers secured with a ratchet strap. All insects were immediately brought back to the Quarantine Lab for immediate use.

### 2.6. Exposing L. delicatula to Symptomatic Trees

On the same day that each of the three *L. delicatula* field collections occurred, an experimental trial was initiated. To do this, five randomly chosen *L. delicatula* were placed into one of 15 pop-up mesh cages 34.3 × 34.3 × 61 cm (13.5 × 13.5 × 24”, BioQuip, 1466BV, bleached and dried thoroughly before use). Ten of these mesh cages contained symptomatic *A. altissima* material while five contained asymptomatic *A. altissima* material. Fourth instars and the first set of adults collected were placed within symptomatic or asymptomatic potted seedling cages, while the second set of adults were exposed to logs collected from symptomatic or asymptomatic trees. The use of logs was included to account for adult *L. delicatula*’s reported preference for feeding on woody plant tissues [3]. These insects were allowed to feed on the provided plant material for 48 h, during which active feeding and insect survival was monitored at the 24 and 48 h marks. Active feeding was confirmed by visually observing each insect and recording the number whose stylet was inserted into the plant material. 

After 48 h of feeding, all live insects were transferred into new pop-up mesh cages containing healthy non-inoculated *A. altissima* seedlings, and allowed to feed for 72 h. The original *A. altissima* material was autoclaved and appropriately disposed of. Active feeding and survival were monitored at 24, 48, and 72 h after the insects were transferred. After 72 h, all the insects were removed from these transfer seedlings.

### 2.7. Culturing V. nonalfalfae from L. delicatula

After removing *L. delicatula* from the feeding trial, each insect was individually placed into a sterile microcentrifuge tube (1.5 ml Fisherbrand^TM^ for fourth instars or 5 ml MacroTube5^TM^ for adults) containing 0.5 mL of sterile deionized (DI) water. These vials were vortexed for 30 sec on high (NuAire, Inc. Plymouth, MN, USA), before the insects were removed using sterile tweezers and appropriately disposed of. The wash water was then plated on the Komada’s agar (500 mL H_2_O, 1 g L-sorborose, 0.5 g K_2_HPO_4_, 0.25 g KCl, 0.25 g MgSO_4_, 0.0005 g EDTA, 1 g asparagine, 8.5 g plain agar, 0.5 g PCNB, 0.25 g oxgall, 0.5 g Na_2_B_4_O, 0.15 g streptomycin), a selective medium for isolating *Verticillium* spp. [22]. At the same time, a known *V. nonalfalfae* culture was streaked onto two control dishes also containing Komada’s agar. All these dishes were stored in the dark at room temperature and checked for Verticillium-like growth after 7 and 14 days. Any Verticillium-like growth observed was transferred and monitored for whirl like conidiophores and the production of resting structures. If Verticillium was morphologically identified, species identification was confirmed using the molecular methods detailed in [23].

### 2.8. Monitoring and Destructively Sampling A. altissima Seedlings

After *L. delicatula* were removed from the final *A. altissima* seedlings, these seedlings were allowed to grow as needed for 21 days in the same laboratory conditions (25 °C, 14 light: 10 dark). They were then assessed for Verticillium wilt-like symptoms and destructively sampled at four locations on each seedling in which *L. delicatula* feeding had been observed (either on the stem or a specific petiole). Each of the areas selected for sampling were surface sterilized using 70% EtOH, cut to a 3 mm cross-section and plated on PEA+SN. Any discoloration of xylem observed was recorded and plated samples were monitored for Verticillium-like growth for 14 days. If Verticillium-like growth was observed, the potential *V. nonalfalfae* isolate was transferred to a new plate and morphologically and molecularly identified to species as described above. After these isolation attempts, all *A. altissima* materials were properly disposed of.

### 2.9. Analysis

The average difference in the number of *V. nonalfalfae* colonies re-isolated in wash water, the number of transfer seedlings displaying Verticillium wilt-like symptoms, the number of *V. nonalfalfae* colonies re-isolated from the transfer seedlings, the *L. delicatula* feeding rate (percent of live insects observed feeding at all observation periods within each cage), and *L. delicatula* survival (the percentage of all insects that survived the five-day experiment within each cage) during each of the three trials was compared using a permutation test run 1000 times. A two-sided *p*-value was calculated by totaling the number of times the absolute value of the null distribution difference of means were equal to or more extreme than the observed differences in means. Any *p*-values less than 0.05 were considered significant. 

## 3. Results

### 3.1. Culturing V. nonalfalfae from L. delicatula

Wash water from 225 *L. delicatula* insects (75 fourth instars and 150 adults) was plated. Monitoring at 7 and 14 days after plating revealed no *V. nonalfalfae*-like growth, even though a positive *V. nonalfalfae* culture plated at the same time and kept in the same environment showed substantial *V. nonalfalfae* growth. 

### 3.2. Culturing V. nonalfalfae from A. altissima

All 45 *A. altissima* seedlings (15 controls and 30 *V. nonalfalfae* treatments) displayed no Verticillium wilt symptoms (leaflet wilt, leaflet chlorosis and necrosis, leaf drop, epicormic sprouting, or mortality) when assessed after three weeks. In addition, when all trees were destructively sampled no stained xylem was observed and no Verticillium-like growth was detected on the plated tissues. 

### 3.3. Active Feeding and Mortality

Active feeding was observed within each cage of five insects on both initial and transfer plant material, with the percentage of live insects observed feeding at all observation periods for each cage ranging from 27–100% (Table 1). For each of the three trials the permutation test comparing treatment indicated that no significant difference in active feeding between treatments existed (Table 1). Though this was not significant, the average feeding was lower on insects exposed to symptomatic seedlings than those exposed to asymptomatic seedlings. This trend was not true for the logs.

At least one insect survived in each cage throughout the five-day experiment, with survival within each cage ranging from 20–100% (Table 1). There were no statistical treatment differences of survival. Though not significant, overall survival within cages provided with symptomatic *A. altissima* material was consistently lower numerically than those provided with asymptomatic *A. altissima* material.

## 4. Discussion

### 4.1. Lycorma delicatula was Unable to Vector V. nonalfalfae between A. altissima in Laboratory Conditions

We found no evidence that *L. delicatula* was able to vector *V. nonalfalfae* from symptomatic to healthy *A. altissima*. Since *L. delicatula* is a phloem feeder [2] it is possible that *L. delicatula* never feeds where *V. nonalfalfae* is found in the xylem. However, *L. delicatula* is known to probe [2], and therefore does have the potential to interact with the xylem. Additionally, electropenetrography (EPG) data for phloem feeding insects, though limited, indicates that xylem feeding is common [24], and has been shown to occur in Fulgoridae [25]. This inability of transferring *V. nonalfalfae* may indicate that *V. nonalfalfae’s* large-scale dissemination might be better influenced by biopesticide applications from land managers, not natural dissemination by insects. Additionally, if *L. delicatula* is unable to vector a vascular wilt pathogen, it may indicate its inability to spread other unwanted plant pathogens. 

We suspect that our results, though a first step in determining vector capabilities of *L. delicatula,* are inconclusive and additional research will be needed to fully understand *L. delicatula*’s vector potential. For example, this experiment was conducted in a highly controlled laboratory environment in which we could ensure that replications were comparable and easily manage *L. delicatula* movement between cages. However, the laboratory conditions were not ideal for the shade intolerant *A. altissima* seedling growth for the extended length of time that these seedlings were monitored. If this experiment was repeated in a field setting using mature trees, not only would *L. delicatula* be provided with a higher quality food source and could be allowed to feed for longer periods of time, but the trees would remain vigorous for longer and, therefore, they could be better monitored for Verticillium wilt disease over much longer periods of time. Although *V. nonalfalfae* symptoms in the field and laboratory conditions typically become visible within 2–3 weeks [13], a smaller initial inoculum amount, such as one theoretically transferred by an insect vector, may extend the time needed to observe symptoms, therefore supporting the need for a longer-term field experiment. 

While we did attempt to confirm the presence of viable *V. nonalfalfae* on the outside of *L. delicatula*, no attempt was made to determine if it was present within the insect. Circulative/propagative transmission of plant pathogens that colonize the vector body have been confirmed in numerous other systems, such as with the phloem-limited phytoplasmas that may be vectored by some Fulgoromorpha [26] and with the cotton fungal pathogen *Ashbya gossypii* [27]. Specific to this system, *E. brandti* was able to pass viable *V. nonalfalfae* propagules in their feces after feeding on infected *A. altissima* materials [19]. Though we did not test *L. delicatula*’s honeydew or their internal tissues for *V. nonalfalfae* presence, our monitoring and destructive sampling of the *A. altissima* seedlings should have identified successful *V. nonalfalfae* transmission regardless of internal or external transmission. If successful transmission is observed in the future, a more detailed study of the location of *V. nonalfalfae* on *L. delicatula* should be completed.

Although we were able to sustain five insects within each cage over the five-day laboratory experiment, testing larger amounts of *L. delicatula* would likely increase the potential for *V. nonalfalfae* to be transferred. This is especially true since numbers of insects counted feeding on a single tree have tallied over 12,000 [13], and therefore if individual *L. delicatula* only rarely acted as a vector, large populations might more consistently transmit disease and more accurately represent *L. delicatula’s* vector capabilities.

### 4.2. Active Feeding Numbers Indicate Method Realistic

Since active feeding was observed in each cage of five insects throughout this experiment, we believe this testing setup within a quarantine facility was effective. Even though there were no statistical differences of feeding rates between the control and *V. nonalfalfae* treated materials in these three trials, cages provided with symptomatic *A. altissima* seedlings tended to have lower feeding rates. Future research determining if *L. delicatula* preferentially fed on asymptomatic or symptomatic seedlings would also help us to better understand the relationship between these three organisms, as effective plant pathogen vectors need to have a high association with diseased plants [28].

### 4.3. Mortality Insignificant between Feeding Treatments

Very high levels of survival were observed during this five-day experiment. This indicates that though *L. delicatula* are not easy to rear in laboratory conditions [3], they can be easily field collected and studied during short periods when their populations are abundant in the field. 

Though not statistically different, no *L. delicatula* insects died when provided with healthy *A. altissima* material, while small portions of fourth instars and adults succumbed when initially provided with symptomatic *A. altissima*. This trend could have implications for *L. delicatula* to sustain a close association with diseased plants which is needed in order to be an effective plant pathogen vector [28]. 

## 5. Conclusions

We found no laboratory support for the ability of *L. delicatula* to vector *V. nonalfalfae* between symptomatic and healthy *A. altissima.* Further testing of *L. delicatula’s* vector ability in the field would help confirm these results.

## Figures and Tables

**Figure 1 insects-11-00515-f001:**
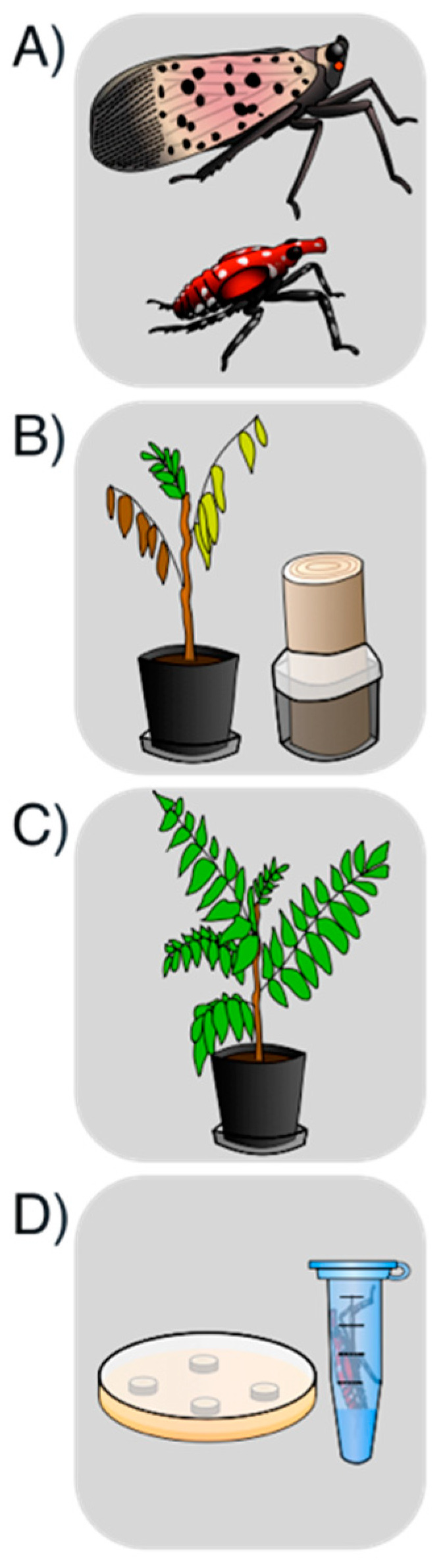
Experimental design illustrated. (**A**) Groups of 5 *L. delicatula* adults or fourth instars were field collected. (**B**) These groups of *L. delicatula* were then placed on *A. altissima* material (either seedlings or logs) that were either symptomatic or asymptomatic and allowed to feed for 48 h. (**C**) These insects were then transferred to healthy non-inoculated seedlings to feed for 72 h before being removed. (**D**) Seedlings were allowed to grow for 3 more weeks before being assessed for symptoms and destructively sampled for *V. nonalfalfae* isolation, while the insects were immediately washed in sterile water that was then plated and monitored for *V. nonalfalfae* growth.

**Figure 2 insects-11-00515-f002:**
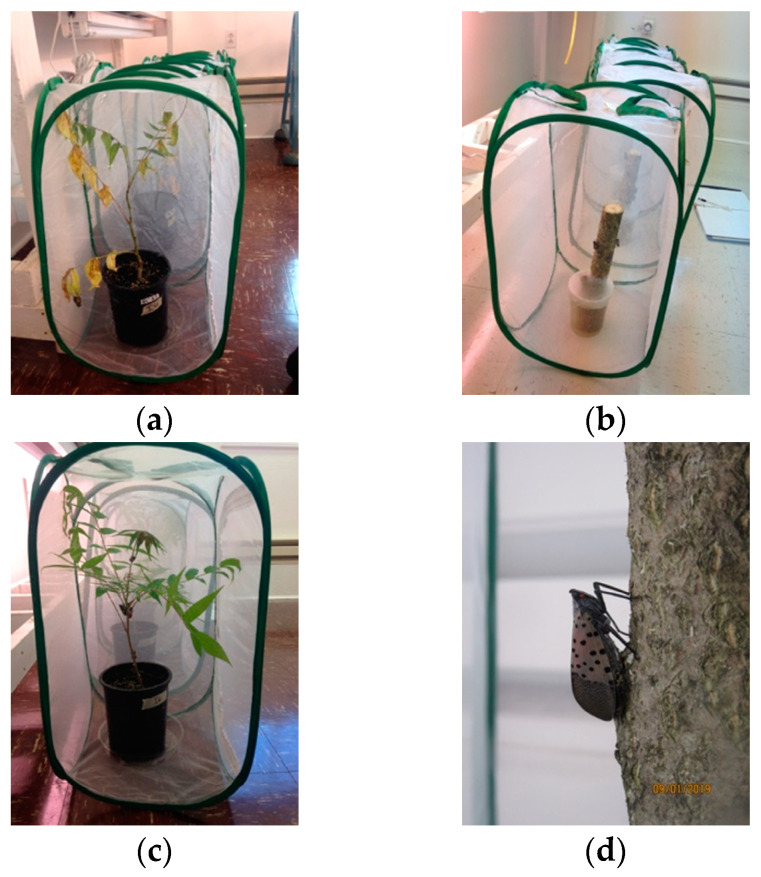
Images from the laboratory experiment: (**a**) *Lycorma delicatula* fourth instars feeding on a symptomatic seedling, (**b**) adults feeding on a healthy log, (**c**) adults feeding on a healthy seedling, (**d**) adult feeding on a symptomatic log, (**e**) adults feeding on a symptomatic seeding, and (**f**) fourth instars feeding on a symptomatic seedling.

**Table 1 insects-11-00515-t001:** The mean, standard deviation (SD), and sample size (*n*) for the percent of live insects observed feeding at all observation periods within each cage (active feeding) and the percent of insects that survived the five-day experiment within each cage (survival). The *p*-values comparing the treatment effect for each trial calculated using a permutation test comparing means are reported for each trial pair.

			Active Feeding (%)				Survival (%)			
Life Stage	Type	Treatment	Mean	SD	*n*	*p*	Mean	SD	*n*	*p*
fourth instar	seedling	control	87.2	7.2	25	0.079	100	0.0	5	0.148
fourth instar	seedling	*V. nonalfalfae*	68.7	20.2	50		74	32.7	10	
adult	seedling	control	90.4	7.8	25	0.194	100	0.0	5	1.000
adult	seedling	*V. nonalfalfae*	85.5	6.2	50		98	6.3	10	
adult	log	control	87.2	15.1	25	1.000	100	0.0	5	0.519
adult	log	*V. nonalfalfae*	87.3	9.3	50		94	9.7	10	

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
