# Peer review of "The Inability of Spotted Lanternfly (Lycorma delicatula) to Vector a Plant Pathogen between its Preferred Host, Ailanthus altissima, in a Laboratory Setting"

_insects, 2020, doi:10.3390/insects11080515_

Round 1

Reviewer 1 Report

This study represents an important area of high interest, as it relates to a potential biological control agent for Ailanthus altissima and relates to the spotted lanternfly (SLF), an important and destructive invasive insect. From their experiments, the authors conclude that SLF is not able to transmit the pathogen Verticillium nonalfalfae among Ailanthus, and go on to state that these findings suggest that SLF may not be able to transmit similar unwanted plant pathogens. As presented, these conclusions are not fully supported by these data. The following describes more specifically the logic for these concerns:

In the current study, SLF are tested for the verticillium after the second feeding by washing the insects in sterile water and plating the wash water to determine if the verticillium grew. Why would you only test the outside of the insect? Couldn't the verticillium be ingested and present in other tissues (salivary glands, gut, etc.)? Does the verticillium need to be incubated within the insect before it can be transmitted? Is 72 hours sufficient time for this to occur? Concerning virus transmission by some hemipterans (e.g., aphids), virus needs to circulate within the insect and there is a huge literature here. While it could very likely be the case that with this kind of fungal pathogen, it is known that the fungus does remain external to the insect and / or that issues of circulation, or issues of the pathogen being harbored in other insect tissues is not relevant at all -- but none of this is touched on in the introduction or the discussion sections. As it stands, it seems that the authors should test for the verticillium within the insect (e.g., salivary glands and / or gut), most likely using molecular methods (e.g., PCR). However, because the Chinese weevil, Eucryptorrhynchus brandti has been studied as a biological control for Ailanthus, it could have been shown with this insect that the pathogen does remain external. Without either doing internal testing of SLF, or using the example of E. brandti to more convincingly show that the verticillium does remain external (i.e., if that literature does exist, it should be discussed), this is a serious limitation of the current study that needs improvement before publication. 

Similar to the above discussion referencing complexities associated with transmitting various plant pathogens (e.g., not only circulating versus noncirculating viruses but also phytoplasmas, etc.), the conclusion that SLF might not transmit other plant pathogens b/c you did not pick up the verticillium on the external surface of SLF in this study is highly misleading and must be rectified before publication. 

Author Response

We thank the reviewer for their comments, and have added a paragraph into the discussion focused on the potential for L. delicatula to transmit V. nonalfalfae internally. We acknowledge that V. nonalfalfae could potentially be vectored both internally and externally by L. delicatula, and agree that this is an important distinction. However, since our main methods involved monitoring and destructively sampling A. altissima seedlings that had been fed upon by L. delicatula which had previously fed on infected A. altissima, we believe that we would have captured the ability of L. delicatula to vector V. nonalfalfae regardless of internal or external transmission. Therefore, with no indication of successful vectoring, further investigation into the details of transmission were not pursued. If, in the future, additional research indicates that L. delicatula can vector V. nonalfalfae from infected to uninfected trees, we believe this area of research should be expanded upon.

Reviewer 2 Report

The manuscript is well written and scientifically sound. The main issue I have is that authors discuss too much results that are not statistically significant, but just trends. I do not like this. If authors are convinced, like they seem to be, that increasing sample size will make those trends significant, they should just go ahead and increase sample size. If they can't/don't want to do this at this time, they should refrain from speculating so much about trends. A brief mention would suffice. 

A couple of minor points:

  • I would specify the fungus name upon first mention (line 48).
  • I am not aware of any phloem feeder that does not have also a xylem feeding phase. Whilst it is true that the number of insects for which EPG data is available is still very small and it's definitely possible that some phloem feeders don't use xylem sap at all, I'd specify that this possibility is, based on current knowledge, not highly likely. Also, I am not sure what authors mean by "overshoot" in the second sentence. Please rephrase the first two sentences of the Discussion.

Author Response

The manuscript is well written and scientifically sound. The main

issue I have is that authors discuss too much results that are not

statistically significant, but just trends. I do not like this. If

authors are convinced, like they seem to be, that increasing

sample size will make those trends significant, they should just

go ahead and increase sample size. If they can't/don't want to

do this at this time, they should refrain from speculating so much

about trends. A brief mention would suffice.

We thank reviewer 2 for their comments. We have attempted to minimize any discussion regarding insignificant results, while still briefly touching upon the trends in order to help structure future research directions.

A couple of minor points:

I would specify the fungus name upon first mention (line 48).

This change was made.

I am not aware of any phloem feeder that does not have also a

xylem feeding phase. Whilst it is true that the number of insects

for which EPG data is available is still very small and it's

definitely possible that some phloem feeders don't use xylem

sap at all, I'd specify that this possibility is, based on current

knowledge, not highly likely.

We have added a sentence in the discussion to highlight this point, thank you for the suggestion.

Also, I am not sure what authors

mean by "overshoot" in the second sentence. Please rephrase

the first two sentences of the Discussion.

We have removed the term “overshoot” and have tried to clarify the first paragraph in the discussion.

Round 2

Reviewer 1 Report

The changes made to the manuscript in light of previous reviewers' comments sufficiently responds to the concerns raised by those reviewers. The manuscript has been improved and now, I recommend, is appropriate for publication in this journal.